# Deep Gradient Boosting – Layer-wise Input Normalization of Neural Networks

## Abstract

Stochastic gradient descent (SGD) has been the dominant optimization method for training deep neural networks due to its many desirable properties. One of the more remarkable and least understood quality of SGD is that it generalizes relatively well on unseen data even when the neural network has millions of parameters. We hypothesize that in certain cases it is desirable to relax its intrinsic generalization properties and introduce an extension of SGD called deep gradient boosting (DGB). The key idea of DGB is that back-propagated gradients inferred using the chain rule can be viewed as pseudo-residual targets of a gradient boosting problem. Thus at each layer of a neural network the weight update is calculated by solving the corresponding boosting problem using a linear base learner. The resulting weight update formula can also be viewed as a normalization procedure of the data that arrives at each layer during the forward pass. When implemented as a separate input normalization layer (INN) the new architecture shows improved performance on image recognition tasks when compared to the same architecture without normalization layers. As opposed to batch normalization (BN), INN has no learnable parameters however it matches its performance on CIFAR10 and ImageNet classification tasks.

## 1 Introduction

Boosting, along side deep learning, has been a very successful machine learning technique that consistently outperforms other methods on numerous data science challenges. In a nutshell, the basic idea of boosting is to sequentially combine many simple predictors in such a way that their combined performance is better than each individual predictor. Frequently, these so called weak learners are implemented as simple decision trees and one of the first successful embodiment of this idea was AdaBoost proposed by Freund & Schapire (1997). No too long after this, Breiman et al. (1998) and Friedman (2001) made the important observation that AdaBoost performs in fact a gradient descent in functional space and re-derived it as such. Friedman (2001) went on to define a general statistical framework for training boosting-like classifiers and regressors using arbitrary loss functions. Together with Mason et al. (2000) they showed that boosting minimizes a loss function by iteratively choosing a weak learner that approximately points in the negative gradient direction of a functional space.

Neural networks, in particular deep neural nets with many layers, are also trained using a form of gradient descent. Stochastic gradient descent (SGD) (Robbins & Monro, 1951) has been the main optimization method for deep neural nets due to its many desirable properties like good generalization error and ability to scale well with large data sets. At a basic level, neural networks are composed of stacked linear layers with differentiable non-linearities in between. The output of the last layer is then compared to a target value using a differentiable loss function. Training such a model using SGD involves updating the network parameters in the direction of the negative gradient of the loss function. The crucial step of this algorithm is calculating the parameter gradients and this is efficiently done by the backpropagation algorithm (Rumelhart et al., 1988; Werbos, 1974).

Backpropagation has many variations that try to achieve either faster convergence or better generalization through some form of regularization. However, despite superior training outcomes, accelerated optimization methods such as Adam (Kingma & Ba, 2015), Adagrad (Duchi et al., 2011) or RMSprop (Graves, 2013) have been found to generalize poorly compared to stochastic gradient

descent (Wilson et al., 2017). Therefore even before using an explicit regularization method, like dropout (Srivastava et al., 2014) or batch normalization (Ioffe & Szegedy, 2015), SGD shows very good performance on validation data sets when compared to other methods. The prevalent explanation for this empirical observation has been that SGD prefers "flat" over "sharp" minima, which in turn makes these states robust to perturbations. Despite its intuitive appeal, recent work by (Dinh et al., 2017) cast doubt on this explanation.

This work introduces a simple extension of SGD by combining backpropagation with gradient boosting. We propose that each iteration of the backpropagation algorithm can be reinterpreted by solving, at each layer, a regularized linear regression problem where the independent variables are the layer inputs and the dependent variables are gradients at the output of each layer, before non-linearity is applied. We call this approach deep gradient boosting (DGB), since it is effectively a layer-wise boosting approach where the typical decision trees are replaced by linear regressors. Under this model, SGD naturally emerges as an extreme case where the network weights are highly regularized, in the $L2$ norm sense. We hypothesize that for some learning problems the regularization criteria doesn't need to be too strict. These could be cases where the data domain is more restricted or the learning task is posed as a matrix decomposition or another coding problem. Based on this idea we further introduce INN, a novel layer normalization method free of learnable parameters and show that it achieves competitive results on benchmark image recognition problems when compared to batch normalization (BN).

## 2 DEEP GRADIENT BOOSTING

Consider a learning problem where training examples $(x, y)$ are generated from a probability distribution $P(x, y)$ where $x \in \mathbb{R}^N$ is the space of measurements and $y \in \mathbb{R}$ is the space of labels. The goal of gradient boosting is to find an approximation $\hat{F}(x)$ to a function $F(x)$ that minimizes a specified loss function $L(F(x), y)$:

$$\hat{F} = \arg\min_F L(F(x), y) \tag{1}$$

Gradient boosting solves Eq. 1 by seeking functions $h_t(x)$, so called weak learners, from a differentiable class $\mathcal{H}$ such that $\hat{F} = \sum_t h_t(x)$. Starting from a constant function $F_0$ and applying steepest descent we can find $\hat{F}$ by iteratively updating with learning rate $\gamma$:

$$F_t(x) = F_{t-1}(x) - \gamma \nabla_{F_{t-1}} L(F_{t-1}(x), y) \tag{2}$$

The weak learners $h_t$ are thus found by fitting the training data to the pseudo-residuals $r_t = \nabla_{F_{t-1}} L(F_{t-1}(x), y)$ hence in practice:

$$F_t(x) = F_{t-1}(x) - \gamma h_t(x) \tag{3}$$

Next, consider a neural network $F(x)$ with parameters $w^{[l]}$ comprised of $L$ successive dense linear layers $F^{[l]}(x) = x^T w^{[l]}$ with non-linearities $\psi$: $F(x) = F^{[0]}(\psi(F^{[1]}(...\psi(F^{[L]}(x)))))$. Similarly, starting from parameter $w_0$ and following steepest descent the network weights are updated at iteration $t$:

$$w_t = w_{t-1} - \gamma \nabla_{w_{t-1}} L(F(x_t), y_t) \tag{4}$$

At each iteration, for a fixed layer $l$ the gradient of the parameter $w_{ij}^{[l]}$ can be calculated using the chain rule:

$$\frac{\partial L_t}{\partial w_{ij}^{[l]}} = \frac{\partial L_t}{\partial F_t^{[l]}} \frac{\partial F_t^{[l]}}{\partial w_{ij}^{[l]}} = \frac{\partial L_t}{\partial F_t^{[l]}} \psi(F_t^{[l-1]})_i = \frac{\partial L_t}{\partial F_t^{[l]}} o_{ti}^{[l]} \tag{5}$$

where $o_i^{[l]}$ is the output of the layer $l-1$ at iteration $t$ and position $i$ and the gradients $\frac{\partial L_t}{\partial F_t^{[l]}}$ are calculated using the backpropagation algorithm.

Alternatively, using the gradient boosting formula from Eq. 3 and plugging in $F_t = F^{[l]}(x_t) = x_t^T w^{[l]}$ for input $x_t = o_t^{[l]}$ we obtain, after dropping the layer indexes $[l]$ for simplicity:

$$o_t^T w_t = o_t^T w_{t-1} + \gamma h(o_t) \tag{6}$$

We choose the weak learner $h(o_t)$ in this case to be a linear transformation as well $h(o_t) = o_t^T v_t$ and fit it to the pseudo-residuals $r_t = \frac{\partial L_t}{\partial F_t^{[l]}}$ by solving the associated regularized linear regression:

$$\hat{v}_t = \arg\min_v (\|r_t - o_t^T v\|_2^2 + \alpha\|v\|_2^2), \alpha > 0 \tag{7}$$

The primal form solution of Eq. 7 is $\hat{v}_t = (o_t^T o_t + \alpha I)^{-1} o_t r_t$ and plugging it into Eq. 6 yields the following update rule: $w_t = w_{t-1} - \gamma(o_t^T o_t + \alpha I)^{-1} o_t r_t$ where $I \in \mathbb{R}^{N \times N}$ is the identity matrix. For a given input batch of size $B$, the update rule becomes:

$$W_t = W_{t-1} - \gamma(O_t^T O_t + \alpha I)^{-1} O_t^T R_t \tag{8}$$

where $W_t \in \mathbb{R}^{n \times m}$ is the weight matrix for a given layer, $O_t \in \mathbb{R}^{B \times n}$ is the input matrix at the same layer while $R_t \in \mathbb{R}^{B \times m}$ is the pseudo-residual matrix $\nabla_{F_t} L_t$. In practice, the inverse of matrix $O_t^T O_t$ can be hard to calculate for neural networks with large number of neurons per layer. In consequence, it is often more convenient to use the dual form formulation of the ridge regression solution of Eq. 7 that involves calculating the inverse of $O_t O_t^T$ instead. This is a well known formulation (e.g. see Chapter 1 of (Camps-Valls et al., 2006)) and leads to the analogous weight update rule:

$$W_t = W_{t-1} - \gamma O_t^T (O_t O_t^T + \alpha I)^{-1} R_t \tag{9}$$

The weight update formulas from Eq. 8 and 9 have the drawback that they change in magnitude with either the number of neurons in the neural network layer or batch size. This is an undesirable behavior that can lead to differences in scale between weights from layers of different sizes. In consequence, the terms $O_t^T O_t$ and $O_t O_t^T$ are scaled accordingly.

For large values of $\alpha$ the terms $O_t^T O_t + \alpha I$ and $O_t O_t^T + \alpha I$ can be approximated by a constant diagonal matrix which leads to the classic SGD solution $W_t \approx W_{t-1} - \gamma O_t^T R_t$. This shows that back-propagation with SGD is an extreme case of DGB that implicitly minimizes the magnitude of weight updates.

## 2.1 EXPERIMENTS

In the following we will compare the performance of DGB and SGD on a variety of reference data sets and explore the effect of the $\alpha$ parameter using simple dense layer neural networks. A summary of the data sets used is given in Table 1:

| Data set | No samples | No features | Task |
|---|---|---|---|
| MNIST | 70,000 | 784 | Image classification |
| Higgs | 1.5M | 28 | Higgs boson event classification |
| Reuters | 11,228 | 1,000 | Newswire topic classification |
| HR | 1470 | 47 | Employee attrition classification |
| Air | 7576 | 12 | Air quality regression |

Table 1: Data sets used in experiments.

The first data set is the MNIST database of handwritten digits (LeCun et al., 1999) comprised of 60,000 28x28 grayscale images for training and 10,000 for testing. In the case of the Higgs data set (Baldi et al., 2014) we used a randomly chosen subset with 1 million samples out of 10 million for training, and 500 thousands samples for testing. This data comes from the field of high energy physics and was generated by Monte Carlo simulations. The third data set (Reuters-21578) is comprised of 11,228 english language news articles from Reuters international news agency, labeled over 46 topics. The forth data set is the Human Resource (HR) data set (Stacker IV, 2015) which was put together by IBM Inc. to predict employee attrition based on 47 features and was randomly split into 1,029 training samples and 441 test samples. The fifth data set is a regression problem for predicting a measure of air quality based on the outputs from an array of chemical sensors together with air temperature and humidity (De Vito et al., 2008).

All experiments were performed until test set convergence using a gradient update momentum of 0.9 and no decay. For simplicity, network architecture was fixed for each data set and all activation

functions were rectified linear units (ReLU) (Nair & Hinton, 2010). The network weights were initialized using a uniform distribution according to (He et al., 2015). The loss functions used were cross-entropy loss for classification problems and mean squared loss for regression problems. The performance metrics used were accuracy for classification tasks and root mean squared error (RMSE) for regression tasks. In addition, standard deviations were calculated and reported for all performance measures by repeating each experiment 10 times with different random seeds.

### 2.1.1 MNIST

For this experiment we found that results were relatively invariant to network architecture passed a certain complexity, in consequence we chose a dense neural network with three hidden layers each with 500 ReLU nodes. All models were trained using 100 sample batch size for 25 epochs using a fixed 0.1 learning rate and achieved 100% accuracy on the training set. The two DGB variants were labeled DGB(l) if it used the left pseudo-inverse from Eq. 8 and DGB(r) corresponding to the right pseudo-inverse from Eq. 9.

| Model | $\alpha$ | Learning rate | Performance |
|-------|-----------|---------------|-------------|
| SGD | | 0.1[0-25] | $98.41\% \pm 0.12$ |
| DGB(r) | 1.0 | 0.1[0-25] | $\mathbf{98.59\%} \pm 0.04$ |
| DGB(r) | 0.1 | 0.1[0-25] | $98.35\% \pm 0.10$ |
| DGB(r) | 0.05 | 0.1[0-25] | $97.99\% \pm 0.11$ |
| DGB(r) | 0.01 | 0.1[0-25] | $96.25\% \pm 0.29$ |
| DGB(l) | 1.0 | 0.1[0-25] | $98.45\% \pm 0.06$ |
| DGB(l) | 0.1 | 0.1[0-25] | $98.25\% \pm 0.12$ |
| DGB(l) | 0.05 | 0.1[0-25] | $98.03\% \pm 0.10$ |
| DGB(l) | 0.01 | 0.1[0-25] | $96.86\% \pm 0.13$ |

Table 2: Performance on the MNIST data set measured as accuracy.

Both left and right variants achieve increasingly better test set performance with larger $\alpha$ values culminating with 98.59% average accuracy for the DGB(r) model with $\alpha = 1.0$. This is marginally better than 98.41% average accuracy for SGD but more importantly this result shows how increasing the regularization parameter alpha successfully improves test set performance for an image classification problem prone to over-fitting.

Overall DGB(l) and DGB(r) show similar performance with the drawback that for relatively small batch sizes DGB(l) is considerably slower. This is because it needs to calculate the inverse of 500x500 matrix as opposed to calculating the inverse of 100x100 matrix, as is the case for DGB(r). When training on a single V-100 Nvidia GPU one SGD iteration took on average 0.0166 sec, one DGB(r) iteration took 0.0188 sec while in contrast DGB(l) took 0.1390 sec. In consequence, for the rest of experiments we will just use DGB(r) and simply label it DGB.

### 2.1.2 HIGGS

Higgs data set was created using Monte Carlo simulations of particle collisions and has 11 mil samples with 28 features. This is a binary classification problem designed to distinguish between a signal process which produces Higgs bosons and a background process which does not. For this exercise we kept only 1 mil samples for training and 500k for testing. All 28 features were normalized to be between 0 and 1. We used an architecture composed of two hidden linear layers with 500 ReLU activations each that was trained using a small batch size of 30 samples over 50 epochs. The learning rate was kept constant at 0.01 for SGD and DGB for high $\alpha$ values and was reduced to 0.001 for $\alpha \leq 0.1$.

This data set poses a considerable challenge with its relatively few features and large sample size. Performance on the training set is relatively poor and doesn't approach 100% accuracy like in the other experiments. The best test set accuracy is obtained for a smaller $\alpha$ value (Table A1).

### 2.1.3 REUTERS

Reuters is a text categorization data set comprised or articles organized into 46 news topics. It was randomly divided into 8982 training and 2246 test samples that were processed using a binary tokenizer for the top 1,000 most common words. This data was then fed into a dense neural network with two hidden layers each with 500 ReLU nodes and trained for 100 epochs with learning rate fixed at 0.01.

Table A2 shows the results for experiments run using the SGD optimizer and DGB with $\alpha$ values 1.0, 0.1 and 0.01. This is a relatively small data set with a large number of features that is usually more successfully addressed using more advanced approaches like recurrent neural networks or convolutional neural nets. In this case, the results are close for all experiments and only marginally better at 78.05% mean accuracy for DGB with $\alpha = 0.1$.

### 2.1.4 HR

Similar to the other data sets, min-max normalization was used on all 47 features of the Human resource attrition data (HR). This is a binary classification task designed to predict employee attrition. To this end we employed a dense neural network with one hidden layer composed of 100 ReLU nodes. As before, we used a batch size of 30 samples for speed. All models were trained for 500 epochs with a fixed learning rate of 0.001.

With only 1029 training samples and 441 test samples the error bars (i.e. standard deviation) are too big to make a claim that DGB outperforms SGD in this case (see Table A3).

### 2.1.5 AIR

The Air quality data set contains 9358 instances of hourly averaged reads from chemical sensors designed to measure air quality. After removing samples with missing values we divided the remaining data equally into 3788 training and 3788 test samples, and all the features were min-max normalized to be between 0 and 1. Unlike the previous experiments this was a regression problem with the goal of predicting concentrations of benzene, a common pollutant associated with road traffic. For this we employed a neural network with two hidden layers with 100 ReLU activations each, trained for 1000 epochs using a fixed learning rate of 0.1 and a batch size of 30 samples.

Just as the previous experiments, DGB with larger $\alpha$ values were closer in performance to SGD (Table A4). In this case, relaxing the regularization parameter led to a gradual increase in test set performance with almost half the root mean squared error for DGB with $\alpha = 0.001$ when compared to SGD.

## 3 FAST APPROXIMATION OF DGB

We showed in the previous section that deep gradient boosting outperforms regular stochastic gradient descent in some cases. However, this comes at a cost because DGB gradient updates involve calculating the inverse of a matrix at each step and for each layer. The computational cost for matrix inversion is roughly $\mathcal{O}(n^3)$ and this can be significant even for small batch sizes as the ones used in the previous experiments. A straight forward way of speeding matrix inversion is to keep just the diagonal terms. This is equivalent of making the simplified assumption that for a layer input matrix $O \in \mathbb{R}^{B \times n}$ the rows are independent (i.e. $\sum_k o_{ik} o_{jk}$ for $i \neq j$) in the case of DGB(r) or the columns are independent (i.e. $\sum_k o_{ki} o_{kj}$ for $i \neq j$) in the case of DGB(l).

For a layer with $n$ input dimensions and a sample batch of size $B$, plugging in the diagonal approximations of matrices $O_t O_t^T$ and $O_t^T O_t$ into Eq. 8 & 9 the gradient updates become:

$$W_t = W_{t-1} - \gamma \hat{O}_t^T R_t \tag{10}$$

where

$$\hat{o}_{i,j} = \frac{o_{ij}}{\frac{1}{n} \sum_{j=1}^n o_{ij}^2 + \alpha} \tag{11}$$

in the case of DGB(r) and

$$\hat{o}_{i,j} = \frac{o_{ij}}{\frac{1}{B}\sum_{i=1}^{B} o_{ij}^2 + \alpha} \qquad (12)$$

in the case of DGB(l).

In the above notation $\hat{O}_t$ is the normalized version of layer input matrix $O_t$ with elements $\hat{o}_{ij}$. In the following paragraphs, FDGB(r) will refer to fast DGB(r) from Eq. 11 and FDGB(l) will refer to fast DGB(l) from Eq. 12.

## 3.1 CONVOLUTIONAL NEURAL NETWORKS

Convolutional neural networks (CNN) are a class of neural networks particularly suited to analyzing visual imagery. Typically the input at each CNN layer is a tensor of size $(B \times c \times w \times h)$ where $B$ is the batch size, $c$ is the number of feature maps while $w$ and $h$ are the width and height of the map. Instead of using a dense network with different weight connecting each point in the $(c \times w \times h)$ map, CNNs work by dividing the feature map into usually overlapping sections by scanning the input with a kernel of a size much smaller than $(w \times h)$, usually $(3 \times 3)$. In this way CNNs effectively increase the number of input samples and simultaneously decrease the number of weights that need to be learned. This process can be viewed as a form of regularization and it has proven to greatly outperform regular dense networks that are usually prone to over-fitting on image recognition problems.

For a given CNN layer, assume we have an input of size $(B \times c_0 \times w_0 \times h_0)$ and an output of size $(B \times c_1 \times w_1 \times h_1)$ that was generated with a kernel of size $(p \times q)$. In this case, the equivalent DGB formulation based on Eq. 8 & 9 would have an input matrix $O_t$ of size $(B \times w_1 \times h_1, c_0 \times p \times q)$. It would be extremely costly to first unfold the input as described and then calculate the matrix inverse of either $O_t^T O_t$ or $O_t O_t^T$. In consequence, it is advisable to use the fast versions of DGB however, one would still have to unfold the input matrix which is still a costly operation.

The key observation for extending FDGB(l) to convolutional neural networks is that for small kernels with a small step size each position in the kernel matrix will "visit" roughly the same pixels in a feature map. Hence, calculating the raw second moment of each feature needs to be done only once per map. After resizing the input tensor $O_t$ to $(B \times w_0 \times h_0, c_0)$ the new normalized input $\hat{O}_t$ becomes:

$$\hat{o}_{i,j} = \frac{o_{ij}}{\frac{1}{B \times w_0 \times h_0}\sum_{i=1}^{B \times w_0 \times h_0} o_{ij}^2 + \alpha} \qquad (13)$$

It is not unreasonable to assume a small kernel size given that most modern convolutional network architectures use kernels of size (3,3) for image recognition.

## 4 INPUT NORMALIZATION

It is clear from Eq. 10 and previous paragraphs that DGB's gradient update formulation can also be viewed as input normalization defined as $N(X) := X(X^T X + \alpha I)^{-1}$ or $N(X) := (XX^T + \alpha I)^{-1}X$ followed by a regular SGD update. Hence this process can also be implemented as a separate normalization layer, similar to how other layer normalization work like batch norm (BN) (Ioffe & Szegedy, 2015), layer norm (LN) (Ba et al., 2016), instance norm (IN) (Ulyanov et al., 2016) and others.

The advantages of these normalization methods are well documented: they accelerate descent by enabling higher learning rates without the risk of divergence which in turn may help avoid local sharp minima (Bjorck et al., 2018), make training more resilient to parameter or input data scale (Ba et al., 2016), and they stabilize layer output variance which prevents gradient exposition in very deep networks (He et al., 2016a). Interestingly, for formulations that operate across samples like batch norm they also have a regularization effect and virtually eliminate the need for other regularization methods like dropout (Srivastava et al., 2014). This is because training examples are seen in conjunction with other examples in the mini-batch, hence the layer outputs are no longer deterministic. This added noise was found to be beneficial to the generalization of the network.

Batch norm, layer norm and instance norm perform a z-score standardization of features/inputs by subtracting the mean and dividing by standard deviation which is typically applied to the output

of a layer, before the activation function. This procedure runs the risk of keeping the output of a layer inside the linear regime of activation functions that are symmetric around zero. In order to recover the expressive power of the network, after normalization is performed two new parameters are introduced that scale and shift the output. These parameters are in turn learned during training like the rest of the network weights. Interestingly, the scale and shift parameters help convergence even for neural networks with non-symmetric activation functions (e.g. ReLU).

In contrast to these methods, the normalization procedure described here is applied to the layer input, it is not scale invariant and has only one parameter $\alpha$ which is predefined before training, not learned. In addition, "normalization" in our case doesn't refer to the process of making the data look more like a standard normal distribution but instead refers to an algebraic transformation of the data that improves its condition number.

## 4.1 INPUT NORMALIZATION OF MULTIPLE LINEAR REGRESSION

In the following we will study the effect of input normalization as defined in the previous paragraph to the simple problem of ordinary least squares solved by gradient descent. Consider a multiple linear regression model $\arg\min_w (y - Xw)^T (y - Xw)$ where $X \in \mathbb{R}^{B \times n}$ is an input matrix, $y \in \mathbb{R}^B$ is the output vector and $w \in \mathbb{R}^n$ is the vector of parameters. It is well known that the rate of convergence of gradient descent in this case depends on the condition number of the Hessian $H = X^T X$ where high values lead to slow convergence. If $X = U \Sigma V^T$ is the singular value decomposition of input matrix $X$, where $U$ and $V$ are orthogonal matrices and $\Sigma$ is a diagonal matrix, then the condition number of $H = X^T X = V \Sigma^2 V^T$ is:

$$\kappa(H) = \frac{\sigma_{max}^2}{\sigma_{min}^2} \tag{14}$$

where $\sigma_{max}$ and $\sigma_{min}$ are the largest and respectively smallest singular values of $X$.

Let $N(X) := X(X^T X + \alpha I)^{-1}$ and $\arg\min_w (y - N(X)w)^T (y - N(X)w)$ be the resulting linear regression problem after applying input normalization. The new Hessian matrix is $H^* = (X^T X + \alpha I)^{-1} X^T X (X^T X + \alpha I)^{-1}$. After plugging in $X = U \Sigma V^T$ and some linear algebra manipulation the new condition number becomes:

$$\kappa(H^*) = \frac{\sigma_{max}^2}{(\sigma_{max}^2 + \alpha)^2} \frac{(\sigma_{min}^2 + \alpha)^2}{\sigma_{min}^2} = \kappa(H) \left( \frac{\sigma_{min}^2 + \alpha}{\sigma_{max}^2 + \alpha} \right)^2 < \kappa(H) \tag{15}$$

This result shows that input normalization leads to a smaller condition number of the Hessian matrix in the case of multiple linear regression. The same result can be obtained when using the alternative input normalization $N(X) := (XX^T + \alpha I)^{-1} X$ (see Prop. A.1 & A.2 in the appendix).

## 4.2 FORMULATION

In practice, we will define Input Normalization (INN) using the faster diagonal approximations from Eq. 11 & 12. In addition, similar to other normalization layers we will also mean center the input data. For a single layer network, centering further improves conditioning of the Hessian of the loss (LeCun et al., 1991), it brings the input data near the critical point of most activation functions thus removing the bias towards certain neurons and, it has a regularization effect by shifting the data in a nondeterministic way during training based on the content of the mini-batch.

For an input matrix $X \in \mathbb{R}^{B,n}$ with elements $x_{ij}$ left input normalization is labeled INN(l) and defined as

$$N(x_{ij}) := \frac{x_{ij} - \frac{1}{B} \sum_{i=1}^{B} x_{ij}}{\frac{1}{B} \sum_{i=1}^{B} x_{ij}^2 + \alpha} \tag{16}$$

while right input normalization is labeled INN(r) and defined as

$$N(x_{ij}) := \frac{x_{ij} - \frac{1}{n} \sum_{j=1}^{n} x_{ij}}{\frac{1}{n} \sum_{j=1}^{n} x_{ij}^2 + \alpha} \tag{17}$$

In the case of INN(l) formulation the batch statistics $m_1 = \frac{1}{B} \sum_{i=1}^{B} x_{ij}$ and $m_2 = \frac{1}{B} \sum_{i=1}^{B} x_{ij}^2$ can cause non-deterministic predictions at test time. Instead, running exponential averages are calculated during training for both $m_1$ and $m_2$. These new values are then used at test time.

### 4.3 EXPERIMENTS

We tested the implicit (i.e. FDGB) and explicit (i.e. INN) input normalization models on the CIFAR10 data set (Krizhevsky et al., 2009) and compared them to Batch Normalization (BN). We then validated the best performing models on the ImageNet object recognition data set (ILSVRC2015) (Russakovsky et al., 2015). In both cases we used versions of the VGG architecture first introduced by Simonyan & Zisserman (2014) and then modified them to accommodate the different models as described in Appendix A.

#### 4.3.1 CIFAR10

The CIFAR10 data set consists of 60000 images organized into 10 classes. There are 50000 training and 10000 test images. According to Table 3 the original VGG11 model achieves a baseline performance of 89.50% accuracy on this problem. Adding batch normalization layers to this architecture significantly improves the performance to 91.18% accuracy. The INN(l) model outperforms both FDGB(l) and VGG11 while matching the performance of batch normalized VGG11 architecture (VGG11_BN). In the next section we validate the performance of INN(l) on a larger image classification data set using a deeper convolutional network.

| Model | $\alpha$ | Learning rate | Performance |
|-------|----------|---------------|-------------|
| VGG11 | | 0.1[0-250], 0.01[250-350] | $89.50\% \pm 0.20$ |
| VGG11_BN | | 0.1[0-250], 0.01[250-350] | $91.18\% \pm 0.16$ |
| FDGB(l) | 1.0 | 0.1[0-250], 0.01[250-350] | $87.91\% \pm 0.30$ |
| FDGB(l) | 3.0 | 0.1[0-250], 0.01[250-350] | $89.10\% \pm 0.25$ |
| INN(l) | 0.5 | 0.1[0-250], 0.01[250-350] | $91.14\% \pm 0.14$ |
| INN(l) | 1.0 | 0.1[0-250], 0.01[250-350] | $\mathbf{91.34\%} \pm 0.18$ |

Table 3: Performance on the CIFAR10 data set measured as accuracy.

#### 4.3.2 IMAGENET

The ImageNet data set is part of the Large Scale Image Recognition Challenge 2015 (Russakovsky et al., 2015) and consists of 1,281,167 training and 50,000 validation images organized into 1,000 categories. For this experiment we compared the performance of VGG19 and ResNet101 with batch normalization (VGG19_BN(a) and respectively ResNet101_BN) to the equivalent architecture with input normalization layers (VGG19_INN(l) and respectively ResNet101_INN(l)) (See Appendix A for more details). In addition, for VGG19_BN(b) we disabled the learnable affine parameters which reduces BN to a basic standardization operation.

| Model | $\alpha$ | Top-5 error | Top-1 error |
|-------|----------|-------------|-------------|
| VGG19_BN(a) | | 6.79% | 23.64% |
| VGG19_BN(b) | | 12.58% | 33.96% |
| VGG19_INN(l) | 1.0 | 6.59% | 23.27% |
| ResNet101_BN | | 5.29% | 20.69% |
| ResNet101_INN(l) | 0.5 | 5.74% | 21.63% |

Table 4: Performance on the ImageNet data set measured as top-1 and top-5 errors.

Similar to CIFAR10 the performance of VGG_INN(l) is very close to that of VGG19_BN(a) although during early epochs VGG_INN(l) shows better convergence (Figure 1). For the BN model without affine parameters (VGG19_BN(b)) performance drops dramatically when trained using the same learning rate schedule showing that extra parameters are essential for the performance of VGG19 with batch normalization (Table 4). When employing an architecture with residual connections ResNet101, replacing batch normalization layers with INN operations leads to a slight decrease in performance.

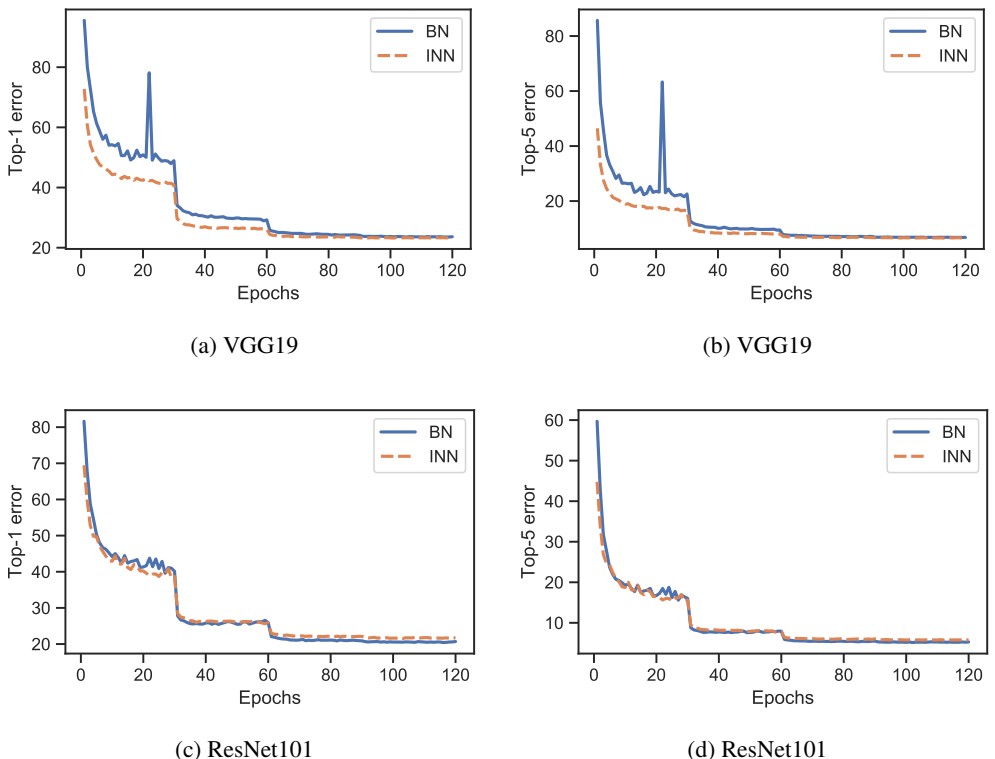

(a) VGG19             (b) VGG19

(c) ResNet101          (d) ResNet101

Figure 1: Top-1 (left) and top-5 (right) validation set errors of VGG_BN(a) and ResNet101_BN labeled as BN and INN(l) labeled as INN over 120 epochs.

## 5 CONCLUSIONS

This work introduces Deep Gradient Boosting (DGB), a simple extension of Stochastic Gradient Descent (SGD) that allows for finer control over the intrinsic generalization properties of SGD. We empirically show how DGB can outperform SGD in certain cases among a variety of classification and regression tasks. We then propose a faster approximation of DGB and extend it to convolutional layers (FDGB). Finally, we reinterpret DGB as a layer-wise algebraic manipulation of the input data and implement it as a separate normalization layer (INN). We then test INN on image classification tasks where its performance proves to be on par with batch normalization without the need for additional parameters.

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

# A  APPENDIX

| Model | $\alpha$ | Learning rate | Performance |
|-------|----------|---------------|-------------|
| SGD   |          | 0.01[0-50]    | $74.10\% \pm 0.12$ |
| DGB   | 1.0      | 0.01[0-50]    | $73.53\% \pm 0.42$ |
| DGB   | 0.1      | 0.001[0-50]   | $74.60\% \pm 0.19$ |
| DGB   | 0.01     | 0.001[0-50]   | **74.99%** $\pm 0.09$ |
| DGB   | 0.001    | 0.001[0-50]   | $73.98\% \pm 1.22$ |

Table A1: Performance on the Higgs data set measured as accuracy.

| Model | $\alpha$ | Learning rate | Performance |
|-------|----------|---------------|-------------|
| SGD   |          | 0.01[0-100]   | $77.54\% \pm 0.29$ |
| DGB   | 1.0      | 0.01[0-100]   | $77.31\% \pm 0.27$ |
| DGB   | 0.1      | 0.01[0-100]   | **78.05%** $\pm 0.23$ |
| DGB   | 0.01     | 0.01[0-100]   | $77.39\% \pm 0.24$ |

Table A2: Performance on the Reuters data set measured as accuracy.

| Model | $\alpha$ | Learning rate | Performance |
|-------|----------|---------------|-------------|
| SGD   |          | 0.001[0-500]  | $86.51\% \pm 0.44$ |
| DGB   | 1.0      | 0.001[0-500]  | $86.87\% \pm 0.21$ |
| DGB   | 0.1      | 0.001[0-500]  | **86.92%** $\pm 0.48$ |
| DGB   | 0.01     | 0.001[0-500]  | $85.24\% \pm 0.59$ |

Table A3: Performance on the HR data set measured as accuracy.

| Model | $\alpha$ | Learning rate | Performance |
|-------|----------|---------------|-------------|
| SGD   |          | 0.1[0-1000]   | $0.0020 \pm 0.00017$ |
| DGB   | 1.0      | 0.1[0-1000]   | $0.0020 \pm 0.00020$ |
| DGB   | 0.1      | 0.1[0-1000]   | $0.0013 \pm 0.00007$ |
| DGB   | 0.01     | 0.1[0-1000]   | $0.0012 \pm 0.00028$ |
| DGB   | 0.001    | 0.1[0-1000]   | $\mathbf{0.0010} \pm 0.00015$ |

Table A4: Performance on the Air data set measured as root mean squared error.

**Proposition A.1.** *If $\sigma$ is a singular value of matrix $X$ then $(X^T X + \alpha I)^{-1} X^T X (X^T X + \alpha I)^{-1}$ has singular values of the form $\frac{\sigma^2}{(\sigma^2 + \alpha)^2}$.*

*Let $X = U \Sigma V^T$ be the singular value decomposition of $X$ then:*

$$
\begin{aligned}
(X^T X + \alpha I)^{-1} X^T X (X^T X + \alpha I)^{-1} &= (V\Sigma^2 V^T + \alpha I)^{-1} V\Sigma^2 V^T (V\Sigma^2 V^T + \alpha I)^{-1} \\
&= (V\Sigma^2 V^T + \alpha V V^T)^{-1} V\Sigma^2 V^T (V\Sigma^2 V^T + \alpha V V^T)^{-1} \\
&= V(\Sigma^2 + \alpha I)^{-1} V^T V\Sigma^2 V^T V(\Sigma^2 + \alpha I)^{-1} V^T \\
&= V(\Sigma^2 + \alpha I)^{-1} \Sigma^2 (\Sigma^2 + \alpha I)^{-1} V^T
\end{aligned}
$$

**Proposition A.2.** *If $\sigma$ is a singular value of matrix $X$ then $X^T (X X^T + \alpha I)^{-2} X$ has singular values of the form $\frac{\sigma^2}{(\sigma^2 + \alpha)^2}$.*

*Let $X = U \Sigma V^T$ be the singular value decomposition of $X$ then:*

$$
\begin{aligned}
X^T (X X^T + \alpha I)^{-2} X &= V\Sigma U^T (U\Sigma^2 U^T + \alpha I)^{-2} U\Sigma V^T \\
&= V\Sigma U^T (U\Sigma^2 U^T + \alpha U U^T)^{-2} U\Sigma V^T \\
&= V\Sigma U^T U(\Sigma^2 + \alpha I)^{-2} U^T U\Sigma V^T \\
&= V\Sigma (\Sigma^2 + \alpha I)^{-2} \Sigma V^T
\end{aligned}
$$

## A.1 CIFAR10

For this experiment we used a version of the VGG11 network introduced by Simonyan & Zisserman (2014) that has 8 convolutional layers followed by a linear layer with 512 ReLU nodes, a dropout layer with probability 0.5 and then a final softmax layer for assigning the classification probabilities. A second version of this architecture (VGG11_BN) has batch normalization applied at the output of each convolutional layer, before the ReLU activation as recommended by Ioffe & Szegedy (2015) We modified this architecture by first removing all the batch normalization and dropout layers. We then either replaced all convolutional and linear layers with ones that implement the fast version of DGB for the FDGB(l) architecture or added INN(l) layers in front of each of the original convolutional and linear layers. Both FDGB(l) and INN(l) models implement input normalization based on the left pseudo-inverse (see Eq. 12 & 16) in order to take advantage of its regularization effect. All weights were initialized according to Simonyan & Zisserman (2014) and were trained using stochastic gradient descent with momentum 0.9 and batch size 128. For the FDGB(l) model the gradients were calculated according to Eq. 12 for linear and 13 for convolutional layers. Training was started with learning rate 0.1 and reduced to 0.01 after 250 epochs and continued for 350 epochs. All experiments were repeated 10 times with different random seeds and performance was reported on the validation set as mean accuracy $\pm$ standard deviation.

## A.2 IMAGENET

Similar to the CIFAR10 data set we based these experiments on the larger VGG19 architecture of Simonyan & Zisserman (2014). The VGG19 network has 16 convolutional layers interspersed with max-pool operations, followed by two linear layers with 4096 ReLU nodes and a final softmax layer with 1000 outputs. The original VGG19 model uses dropout regularization after each rectified linear

layer. A second version of this architecture (VGG19_BN) has batch normalization applied at the output of each convolutional layer, before the ReLU activation. We tested VGG19_BN with and without affine parameters (VGG19_BN(a) and respectively VGG19_BN(b)). After removing all the dropout and batch normalization layers we created the INN(l) version by adding input normalizations in front of each remaining layer.

In order to explore the performance of INN on network architectures with residual connections we employed ResNet101 proposed by He et al. (2016b). The original network uses batch normalization between convolutional layers by default (ResNet101_BN). The INN version of this network (ResNet101_INN(l)) was created by first removing all BN layers, then adding INN operations in front of each convolutional and dense layers, with the exception of down-sample convolutions.

Stochastic gradient descent was used to minimize the cross entropy loss of each network over 120 epochs, using an initial learning rate of 0.05, 0.9 momentum, a batch size of 128 images, and 0.0001 weight decay. Every 30 epochs the learning rate was decreased by a factor of 10. All networks were evaluated by computing the top-1 and top-5 validation errors on a held out data set using 10 crops per image.

