# OpenReview forum: "Deep Gradient Boosting -- Layer-wise Input Normalization of Neural Networks"
_ICLR.cc/2020/Conference — Reject_

### Official Review · AnonReviewer3 · 2019-10-22
**Official Blind Review #3**

**Rating:** 3

**Review:**

Summary
This paper propose an extention method of SGD, deep gradient boosting (DGB), which views the back-propagation procedure as a pseudo-residual targets of a gradient boosting problem. To apply DGB to the real CNNs, DGB is simplified to a input normalization layer, conditioned on the assumption that the convolution kernels should be small. After applying the input normalization layer to CNNs, the model could achieve comparable performance to the model with BN on CIFAR-10 and ImageNet recognition.

There are several concerns influencing my rating:
* I cannot catch the advantages of this input normalization layer compared to BN. For example, could this input normalization layer help to address the problem that BN performs bad when batch size is small? The authors mention that this layer does not have additional parameters. But as I know, the parameter size of BN is small, which downgrades the significance of the proposed method.

* In the CIFAR-10 and ImageNet experiments, only the VGG model is adopted, which obviously limits the application scope. Could the proposed method work well on ResNet, DenseNet or other more recent deep architectures?

* In the DGB experiments, the improvements of DGB compared with SGD in four datasets all seem marginal, in which DGB is slower than SGD.

Overall, I recognize the exploration of this method. But the advantages of DGB compared to SGD seem marginal.


**Experience Assessment:**

I have published in this field for several years.

**Review Assessment: Checking Correctness Of Derivations And Theory:**

I assessed the sensibility of the derivations and theory.

**Review Assessment: Checking Correctness Of Experiments:**

I carefully checked the experiments.

**Review Assessment: Thoroughness In Paper Reading:**

I read the paper at least twice and used my best judgement in assessing the paper.

---

> ### Author Response · Authors · 2019-11-15
> **Thank you**
>
> Thank you for carefully reading the paper.
>
> Though we mostly focus on INN(l), INN(l) is an alternative to batch norm and INN(r) is an alternative to layer norm. This work shows how both formulations are equivalent and emerge from the dual solution to ridge regression. INN also gives a specific meaning to alpha which can be treated as a hyper-parameter and adjusted according to the problem. It is true that the parameter space of BN is small in comparison the the space of parameters of modern deep networks but could be significant in smaller applications. In addition, without the extra parameters memory saving schemes could be employed where the intermediate outputs don't need to be saved during the forward pass and could be calculated on the fly during the backward pass.
>
> For ImageNet, we have added a couple of experiments performed using ResNet101 to the manuscript.
>
> We have added running times for the MNIST experiment and showed that for a batch size of 100 samples DGB(r) takes just 13% more time per iteration. We'd also like to point out that the performance increase for Air and Higgs are significant given the standard deviations shown in Tables A1 and A4.

---

### Official Review · AnonReviewer1 · 2019-10-22
**Official Blind Review #1**

**Rating:** 3

**Review:**

The paper proposes an a new idea of treating the back propagated gradients using chain rule as pseudo residual targets of a gradient boosting problem. Then the weight update is done by solving the boosting problem using a linear base learner. Furthermore, to reduce computational cost incurred by solving the boosting problem, an idea proposed is only keep the diagonal terms of the matrix inversion involved.

It is an interesting idea to look at gradients different and update weight accordingly. The experimental results shows some improvements.

The paper is written in a way that it looks like two papers and connected with the same idea DGB. It is a bit confusing during the first pass of reading.

In all experiments, can you also report running times? how much is the overhead? and  how much reduction in training time if only using diagonal terms?

Does only using diagonal terms hurt model performance?

For 2.1.1 MNIST, why is alpha the larger the better while the others are not this case?

For 2.1.5 AIR, why spit training and test equality?

For 4.3 why choose heavy model VGG? not ResNet?

In Figure 1, is the spike of BN normal?



**Experience Assessment:**

I have read many papers in this area.

**Review Assessment: Checking Correctness Of Derivations And Theory:**

I assessed the sensibility of the derivations and theory.

**Review Assessment: Checking Correctness Of Experiments:**

I assessed the sensibility of the experiments.

**Review Assessment: Thoroughness In Paper Reading:**

I read the paper at least twice and used my best judgement in assessing the paper.

---

> ### Author Response · Authors · 2019-11-15
> **Thank you**
>
> We appreciate taking the time to review our paper. Bellow are our answers, one paragraph for each comment.
>
> We added running times for the MNIST experiment in the manuscript: "When training on a single V-100 Nvidia GPU one SGD iteration took on average 0.0166 sec, one DGB(r) iteration took 0.0188 sec while in contrast DGB(l) took 0.1390 sec." With a small enough batch size, in this case 100 samples, the overhead is small, about 13% increase in running time. When using just the diagonal elements one training iteration takes 0.0175 sec.
>
> The diagonal approximation doesn't perform as well as the full model and we use it just for the CIFAR10 experiments, where the convolutional version of DGB does not accept the full model. However, considering that the computational overhead is not too big its worth using the full model for dense networks with relatively small batch sizes.
>
> The ideal alpha value varies from problem to problem, similar to how lamda values differ in the case of ridge regression. MNIST is highly susceptible to overfitting when using dense networks thus more regularization (ie larger alpha) helps. In other case, like Higgs where we have a lot of samples and very few features, the main issue is convergence not overfitting, thus smaller alphas work better.
>
> As a regression problem AIR has a bigger range of values that need to be fit, thus a larger test set that is more representative is recommended.
>
> We initially choose VGG because it lacked residual connections. Without residual connections DGB is easy to formulate and solve. Nonetheless, in the end input normalization (INN) can be applied to any architecture and we included in the manuscript results on ImageNet using ResNet101.
>
>  A spike like that can happen in the early epochs if SGD finds a local sharp minima, but it quickly exited it.

---

### Official Review · AnonReviewer2 · 2019-10-23
**Official Blind Review #2**

**Rating:** 3

**Review:**

The paper presents a new gradient update rule for neural net weights, which is a result of relaxing the usual SGD to a regularized linear regression between the descent direction and the original gradient. The method is interpreted as some kind of input normalization similar to Batch Norm later in the paper.

Although the method seems to work and the idea is interesting, I’m not entirely convinced by the experimental evidence. There are some technical concerns:

- In section 2.1, I’m only familiar with the MNIST dataset. Although the proposed method achieves slightly better performance than vanilla SGD, it’s pretty weird the accuracy is only 98.xx% as a very simple CNN could easily achieve >99% accuracy on MNIST. This might be due to the simple network architecture used. A more modern architecture would be preferable in those experiments. Also the improvement on CIFAR seems to close to tell.

- The writing seems to need some improvement. It’s sometimes hard to follow the text, and some result tables (Table A1,A2,A3) are in the appendix not the main text. It might be better to reorganize the text and present everything about the method (fast approximation, CNN, interpretation as normalization) together.

- The authors name the proposed method “deep gradient boosting”, but I’m not entirely sure how it is related to gradient boosting. Eq. 3 concerns sum of weak predictors, but how is this related to Eq. 4? The neural net F(x) doesn’t seem to be exactly a sum of weak learners. I don’t quite follow Eq. 5 – There seems to be missing something.

Typos:
- Eq. 5, d L_t instead of d L_T, and should there be a d \phi / d w_ij term?
- Eq. 7, v_t => v on the right-hand side.
- Eq. 12, should be \sum_i=1^B


**Experience Assessment:**

I have read many papers in this area.

**Review Assessment: Checking Correctness Of Derivations And Theory:**

I carefully checked the derivations and theory.

**Review Assessment: Checking Correctness Of Experiments:**

I assessed the sensibility of the experiments.

**Review Assessment: Thoroughness In Paper Reading:**

I read the paper at least twice and used my best judgement in assessing the paper.

---

> ### Author Response · Authors · 2019-11-15
> **Thanks**
>
> We thank the reviewer for the comments.
>
> Regarding the MNIST results, we purposely used a dense network that is known to overfit on image classification problems in order to show the regularization effects of the alpha parameter. One can observe how for both left and right formulations the test set accuracies improve. We do not make a point that DGB(r) with alpha=1 is significantly better than the SGD results given the provided standard deviations. We do point in the subsequent experiments that having a finer control (through alpha) of SGDs intrinsic regularization properties can be beneficial. For example, the performance on the Air data set is 2x better than SGD with very small error bars. CIFAR10 results also point out towards the regularization effects of alpha but also tests the extensions of DGB to convolutional networks and layer normalization procedures. In this and subsequent experiments we show how input normalization (INN) converges faster than batch normalization (BN) and ultimately the results are very close. Thus INN presents itself as a viable alternative to BN and because it doesn’t have learnable parameters it is more light weight.
>
> Unfortunately adding the tables in the main text would break the 8 page limit, this is the only reason they are in the appendix.
> The main inspiration for this work comes from stochastic gradient boosting. The derivation described in section 2 shows how we can modify backprop to perform a gradient descent in functional space. With backprop first the gradients at each layer are calculated using the chain rule. Now instead of using them to calculate directly the gradients of the parameters w via o*r we can directly estimate the functional gradients r from the output of the respective linear layer. Solving this linear equation then gives us the weight update rule. Plugging in F = ow and h = ov in Eq 3 gives the weight update corresponding to the one in Eq 4 but obtained via gradient boosting. The weak learners F_t are summed in time; at each iteration we determine F_t = w_t*x_t and then use the weights gamma*w_t to update the parameter w.
>
> Thanks for noticing, we have corrected to dL_t. Since F = o * w, dF/dw is o which is the output from the previous layer after the non-linearity.
>
> Thank you, we have switched to v in the right hand side of Eq 7
>
> Thank you, we have fixed the sum expression in Eq 12.

---

### Public Comment · ~Boris_Ginsburg1 · 2019-10-06
**Typo**

Table 4: top-1 and top-5 are swapped in the table header.

---

> ### Author Response · Authors · 2019-10-10
> **Thank you for catching this**
>
> Thank you Boris. we will correct this.

---

### Decision · Program_Chairs · 2019-12-19

**Decision:**

Reject

**Comment:**

The paper introduces a neat idea that an SGD update can be written as a solution of the linear least squares problem with a given backpropagated output; this is generalized to a larger batch size, giving a sort of "block" gradient-type update. Some notes that the columns of $O_t$ have to be scaled are made, but not clear why. The paper then goes into the experiments, and then gets back to the fast approximation of DGB. It really looks like bad organization of the paper, which was noted.
The reviewers agree that the actual computational improvements are marginal, and all recommend rejection. As a recommendation, I would suggest to restructure the paper for a more coherent view, and also the improvements in Top-1 are not very stimulating. The general view is interesting, but it is not clear what insight it brings.